# Supervised dimensionality reduction for big data

Joshua T. Vogelstein [1,2✉], Eric W. Bridgeford[1,2], Minh Tang[1], Da Zheng[1], Christopher Douville[1], Randal Burns[1] & Mauro Maggioni[1]

To solve key biomedical problems, experimentalists now routinely measure millions or billions of features (dimensions) per sample, with the hope that data science techniques will be able to build accurate data-driven inferences. Because sample sizes are typically orders of magnitude smaller than the dimensionality of these data, valid inferences require finding a low-dimensional representation that preserves the discriminating information (e.g., whether the individual suffers from a particular disease). There is a lack of interpretable supervised dimensionality reduction methods that scale to millions of dimensions with strong statistical theoretical guarantees. We introduce an approach to extending principal components analysis by incorporating class-conditional moment estimates into the low-dimensional projection. The simplest version, Linear Optimal Low-rank projection, incorporates the class-conditional means. We prove, and substantiate with both synthetic and real data benchmarks, that Linear Optimal Low-Rank Projection and its generalizations lead to improved data representations for subsequent classification, while maintaining computational efficiency and scalability. Using multiple brain imaging datasets consisting of more than 150 million features, and several genomics datasets with more than 500,000 features, Linear Optimal Low-Rank Projection outperforms other scalable linear dimensionality reduction techniques in terms of accuracy, while only requiring a few minutes on a standard desktop computer.

[1] Johns Hopkins University, Baltimore, MD, USA. [2]These authors contributed equally: Joshua T. Vogelstein and Eric W. Bridgeford. ✉email: jovo@jhu.edu

Supervised learning—the art and science of estimating statistical relationships using labeled training data—has enabled a wide variety of basic and applied findings, ranging from discovering biomarkers in omics data[1] to recognizing objects from images[2]. A special case of supervised learning is classification, where a classifier predicts the "class" of a novel observation (for example, by predicting sex from an MRI scan). One of the most foundational and important approaches to classification is Fisher's Linear Discriminant Analysis (LDA)[3]. LDA has a number of highly desirable properties for a classifier. First, it is based on simple geometric reasoning: when the data are Gaussian, all the information is in the means and variances, so the optimal classifier uses both the means and the variances. Second, LDA can be applied to multiclass problems. Third, theorems guarantee that when the sample size $n$ is large and the dimensionality $p$ is relatively small, LDA converges to the optimal classifier under the Gaussian assumption. Finally, algorithms for implementing it are highly efficient.

Modern scientific datasets, however, present challenges for classification that were not addressed in Fisher's era. Specifically, the dimensionality of datasets is quickly ballooning. Current raw data can consist of hundreds of millions of features or dimensions; for example, an entire genome or connectome. Yet, the sample sizes have not experienced a concomitant increase. This "large $p$, small $n$" problem is a non-starter for many classical statistical approaches because they were designed with a "small $p$, large $n$" situation in mind. Running LDA when $p \geq n$ is like trying to fit a line to a point: there are infinitely many equally good fits (all lines that pass through the point), and no way to know which of them is "best". Therefore, without further constraints these algorithms will overfit, meaning they will choose a classifier based on noise in the data, rather than discarding the noise in favor of the desired signal. We also desire methods that can adapt to the complexity of the data, are robust to outliers, and are computationally efficient. Several complementary strategies have been pursued to address these $p \geq n$ problems.

First, and perhaps the most widely used method, is Principal Components Analysis (PCA)[4]. According to PubMed, PCA has been referenced over 40,000 times, and nearly 4000 times in 2018 alone. This is in contrast to other methods that receive much more attention in the media, such as deep learning, random forests, and sparse learning, which received ~2000, ~1200, and ~500 hits, respectively. This suggests that PCA remains the most popular workhorse for high-dimensional problems. PCA "preprocesses" the data by reducing its dimensionality to those dimensions whose variance is largest in the dataset. While highly successful, PCA is a wholly unsupervised dimensionality reduction technique, meaning that PCA does not use the class labels while learning the low-dimensional representation, resulting in suboptimal performance for subsequent classification. Nonlinear manifold learning techniques generalize PCA[5], but also typically do not incorporate class label information; moreover, they scale poorly. Deep learning provides the most recent version of nonlinear manifold learning, for example, using (supervised) autoencoders, but these methods remain poorly understood, have many parameters to tune, and typically do not provide interpretable results[6]. Further, deep learning tends to suffer in the wide data problem, where the number of samples is far less than the dimensionality.

The second set of strategies regularize or penalize a supervised method, such as regularized LDA[7] or canonical correlation analysis (CCA)[8]. Such approaches can drastically overfit in the $p > n$ setting, tend to lack theoretical support in these contexts, and have multiple "knobs" to tune that are computationally taxing. Partial least squares (PLS) is another popular method in this set that often achieves impressive empirical performance, though it lacks strong theoretical guarantees and a scalable implementation[9,10]. Sparse methods are the third common strategy to mitigate this "curse of dimensionality"[11–13]. Unfortunately, exact solutions are computationally intractable, and approximate solutions have theoretical guarantees only under very restrictive assumptions, and are quite fragile to those assumptions[14]. Thus, there is a gap: no existing approach can classify multi-class wide data with millions of features while obtaining strong theoretical guarantees, favorable and interpretable empirical performance, and a flexible, robust, and scalable implementation.

To address these issues, we developed a technique for incorporating class-conditional moment estimates, XOX, the simplest example of which is LOL. The key intuition behind LOL is that we can jointly use the means and variances from each class (like LDA and CCA), but without requiring more dimensions than samples (like PCA), or restrictive sparsity assumptions. Using random matrix theory, we are able to prove that when the data are sampled from a Gaussian, LOL finds a better low-dimensional representation than PCA, LDA, CCA, and other linear methods. Under relatively relaxed assumptions, this is true regardless of the dimensionality of the features, the number of samples, or the number of dimensions in which we project. We then demonstrate the superiority of techniques derived using the XOX approach—including (i) LOL, (ii) a variant of XOX which allows greater flexibility of the class-conditional covariances called QOQ, and (iii) a robust variant of LOL called RLOL—over other methods numerically on a variety of simulated settings including several not following the theoretical assumptions. Finally, we show that on several 500 gigabyte neuroimaging datasets, and several multi-gigabyte genomics datasets, LOL achieves superior accuracy at lower dimensions while requiring only a few minutes of time on a single workstation.

## Results

**Flexibility and accuracy of XOX framework.** We empirically investigate the flexibility and accuracy of XOX using simulations that extend beyond theoretical claims. For three different scenarios, we sample 100 training samples each with 100 features; therefore, Fisher's LDA cannot solve the problem (because there are infinitely many ways to overfit). We consider a number of different methods, including PCA, rrLDA, PLS, random projections (RP), and CCA to project the data onto a low dimensional space. After projecting the data, we train either LDA (for the first two scenarios) or quadratic discriminant analysis (QDA, for the third scenario), which generalizes LDA by allowing each class to have its own covariance matrix[15]. For each scenario, we evaluate the misclassification rate on held-out data.

Figure 1 shows a two-dimensional scatterplot (left) and misclassification rate versus dimensionality (right) for each simulation. Hereafter, LOL will refer to the version of LOL with a robust estimate of the location (the class medians, related to the central moment when the population has a symmetric distribution), and a truncated singular value decomposition to estimate of the second moment. A robust location estimate tends to make little difference when a robust estimate was not necessary, and empirically improves performance in simulations and real-data examples when a robust estimate was warranted. Alternative strategies would have been to use robust estimates of the first moment or second moment directly[16–18]. We do not use a robust estimate of the second moment, as typical robust estimates of the second moment available in standard numerical packages require $d < n$, which is unsuitable for wide data. The top $C - 1$ embedding dimensions for LOL correspond to the performance after projection onto the class-conditional means, and rrLDA

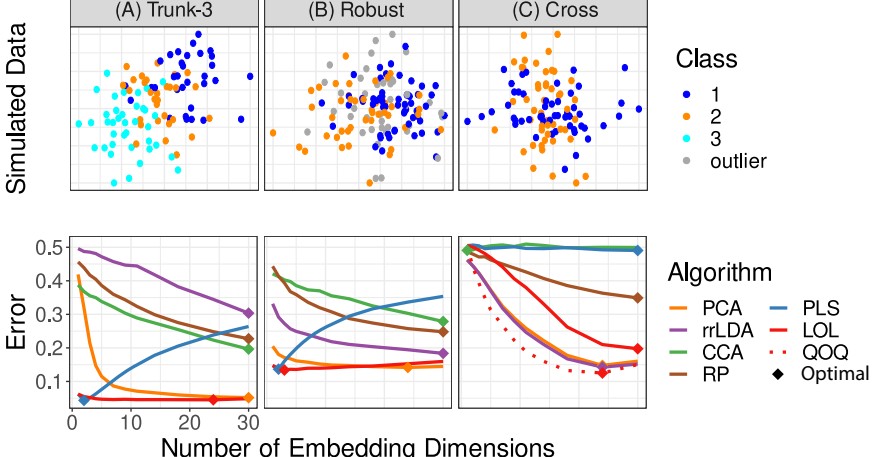

**Fig. 1 Three simulations demonstrating the flexibility and accuracy of** XOX **in settings beyond current theorical claims.** For all cases, training sample size and dimensionality were both 100. The top row depicts the values of the sampled points for two of the 100 dimensions to illustrate the classification task. The bottom row misclassification rate as a function of the number of projected dimensions, for several different embedding approaches. Classification is performed on the embedded data using the LDA classifier for (**A**) and (**B**), and using QDA for (**C**). The simulation settings are: **A** Trunk-3 A variation of Fig. 5b in which three classes are present. **B** Robust Outliers are prominent in the sample while estimating the projection matrix. LOL is robust to the outliers due to the robust estimate of the first moment. **C** Cross The two classes have the same mean but orthogonal covariances. Points are classified using the QDA classifier after projection. QOQ, a variant of LOL where each class' covariance is incorporated into the projection matrix, outperforms other methods, as expected. In essentially all cases and dimensions, LOL, or the appropriate generalization thereof, outperforms other approaches.

corresponds to the performance of projection onto the class-conditional covariance matrix. Figure 1a shows a three class generalization of the Trunk example from Fig. 5b. LOL can trivially be extended to more than two classes (see Supplementary Note 2 for details), unlike ROAD which only operates in a two-class setting. Figure 1b shows a two-class example with many outliers, as is typical in modern biomedical datasets. Both LOL and PLS perform well, despite the outliers, and efficiently identify embedding dimensions despite the outliers. Figure 1c shows an example which should be adversarial for LOL in comparison to PCA or rrLDA. This is because the difference of means is utterly informative, so LOL utilizes additional dimensions which are noise compared to PCA. Further, the class-conditional covariances are orthogonal, whereas LOL assumes the class-conditional covariance is the same across both classes. While LOL cannot possibly do as well as PCA in this situation, its performance is only slightly worse. Further, another XOX variant, quadratic optimal QDA (QOQ), uses the same difference of means as LOL and then computes the eigenvectors separately for each class, concatenates them (sorting them according to their singular values), and then classifies with QDA instead of LDA. QOQ is able to identify a slightly more efficient projection for classification than PCA. This is due to the fact that while the first few dimensions are uninformative (those spanned by the difference of the means), the successive dimensions are far more efficient (the class-conditional covariances). For all three scenarios, either LOL —or its extended variant QOQ—achieves a misclassification rate comparable to or lower than other methods, for all dimensions. These three results demonstrate how straightforward generalizations of LOL under the XOX framework which incorporate alternate or robust moment estimates can dramatically improve performance over other projection methods. This is in marked contrast to other approaches, for which such flexibility is either not available, or otherwise problematic.

**XOX is computationally efficient and scalable.** When the dimensionality is large (e.g., millions or billions), the main bottleneck is sometimes merely the ability to run anything on the data, rather than its predictive accuracy. We evaluate the computational efficiency and scalability of LOL in the simplest setting: two classes of spherically symmetric Gaussians (see Supplementary Note 3 for details) with dimensionality varying from 2 million to 128 million, and 1000 samples per class. Because LOL admits a closed form solution, it can leverage highly optimized linear algebra routines rather than the costly iterative programming techniques currently required for sparse or dictionary learning type problems[19]. To demonstrate these computational capabilities, we built FlashLOL, an efficient scalable LOL implementation with R bindings, to complement the R package used for the above figures.

Four properties of LOL enable its scalable implementation. First, LOL is linear in both sample size and dimensionality (Fig. 2a, solid red line). Second, LOL is easily parallelizable using recent developments in "semi-external memory"[20–22] (Fig. 2a, dashed red line demonstrates that LOL is also linear in the number of cores). Also note that LOL does not incur any meaningful additional computational cost over PCA (orange dashed line). Third, LOL can use randomized approximate algorithms for eigendecompositions to further accelerate its performance[23,24] (Fig. 2a, orange lines). FlashLFL, short for Flash Low-rank Fast Linear embedding, achieves an order of magnitude improvement in speed when using very sparse RP instead of the eigenvectors. Fourth, hyper-parameter selection for LOL is nested, meaning that once estimating the $d$-dimensional projection, every lower dimensional projection is automatically available. This is in contrast to tuning the weight of a penalty term, which leads to a new optimization problem for each different parameter values. Thus, the computational complexity of LOL is $\mathcal{O}(npd/Tc)$, where $n$ is sample size, $p$ is the dimension of the data, $d$ is the dimension of the projection, $T$ is the number of threads, and $c$ is the sparsity of the projection.

Finally, note that this simulation setting is ideal for PCA and rrLDA, because the first principal component includes the mean difference vector. Nonetheless, both LOL and LFL achieve near optimal accuracy, whereas rrLDA is at chance, and PCA requires 500 dimensions to even approach the same accuracy that LOL achieves with only one dimension. While PCA would also benefit efficiency wise from a randomized approach, we emphasize that

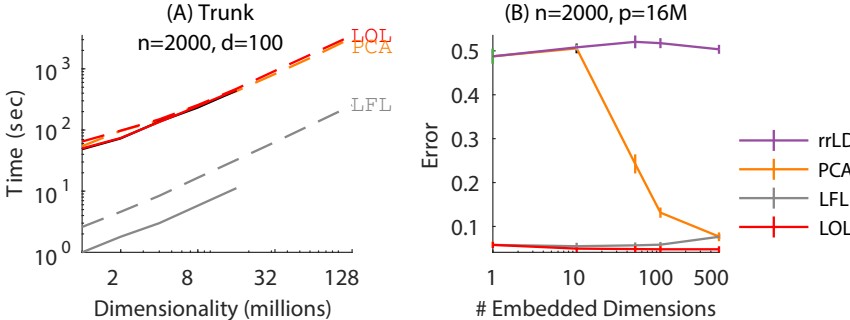

**Fig. 2 Computational efficiency and scalability of** `LOL` **using** *n* = **2000 samples from spherically symmetric Gaussian data (see Supplementary Note 3 for details). A** `LOL` exhibits optimal (linear) scale up, requiring only 46 min to find the projection on a 500 gigabyte dataset, and only 3 min using `LFL` (dashed lines show semi-external memory performance). **B** Error for `LFL` is the same as `LOL` in this setting, and both are significantly better than `PCA` and `rrLDA` for all choices of projection dimension, regardless of whether a randomized approach is used to compute the projection dimensions. Note that while similar scalability enhancements can be made to `PCA` in (**A**), our focus is to highlight that `LFL` maintains the high performance of `LOL` in comparison to `PCA` in (**B**) despite the randomization technique.

`LFL` maintains the high performance of `LOL` in comparison to `PCA` despite the randomization technique, with the benefit of greater computational efficiency compared to `LOL`.

**Real data benchmarks and applications**. Real data often break the theoretical assumptions in more varied ways than the above simulations, and can provide a complementary perspective on the performance properties of different algorithms. We describe two sets of problems, one from brain imaging, and the other from genomics. In both cases we consider a classification problem. To classify participants, researchers typically employ substantiative preprocessing pipelines[25] to reduce the dimensionality of the data. Unfortunately, as debates persist about the validity of pre-processing approaches, there is no defacto "standard" for the optimal strategies to preprocess the data. Traditional approaches typically include a deep processing chain, with many steps of parametric modeling and downsampling[26–28]. We therefore investigate the possibility of directly classifying on the nearly raw, high-dimensional data.

The Consortium for Reliability and Reproducibility (CoRR)[29] has generated anatomical and diffusion magnetic resonance imaging scans from *n* > 800 participants from five processing sites, each featuring participant-specific annotations for the sex of each individual. At the native resolution, each brain volume is over 150 million dimensions, and each dataset consists of between 42 (60 GB of data) and >400 samples (600 GB of data).

We then also consider a large genomics dataset[30] consisting of 340 individuals: 144 patients with nonmetastatic cancer and 196 healthy controls, of which 198 are male and 142 are female. Samples are aligned to > 750,000 amplicons distributed throughout the genome to investigate the presence of aneuploidy (abnormal chromosomal counts) in samples from cancer patients (see Supplementary Note 5 for details). The raw amplicon counts are then used with no further preprocessing. We have two tasks of interest: classification on the basis of either sex or age.

For each of the above described problems, we first compute an embedding matrix to project the training data using `LOL`, `PCA`, `rrLDA`, and `RP`, and then train `LDA` to classify the resulting low-dimensional representations. The held-out set is then projected and classified using the embedding matrix and trained classifier respectively, and the average cross-validated error is computed over all folds of the data. For each problem, the optimal dimensionality for each strategy is selected to be the number of embedding dimensions with the lowest average cross-validated error. We compute Cohen's Kappa $\kappa$ to compare performance across methods because it normalizes the performance of the

classification strategy between zero (the classifier is equivalent to the `random chance` classifier) and one (the classifier performs perfectly). Finally, for each projection technique, we measure the effect size for each strategy as the difference $\kappa(\text{PCA}) - \kappa(\text{embed})$. See Supplementary Table 1 for a table detailing the datasets employed.

Our `FlashLOL` implementations are the only algorithms that could successfully run on these data with a single core on a standard desktop computer. In Fig. 3a, `LOL` is the only technique to outperform `PCA` on all problems. Figure 3b shows the relative ranks of the average cross-validated misclassification rates for the `LDA` classifier on each dataset after projection with the specified embedding technique. For all problems, `LOL` is the technique with the lowest average cross-validated misclassification rate. Further, `LOL` performs significantly better than all other techniques (Wilcoxon signed-rank statistic, all *p* values = 0.008). The average misclassification rate achieved at the optimal number of embedding dimensions via `LOL` is between 5% and 15% across all datasets, which is the same performance we and others obtain using extensively processed and downsampled data that is typically required on similar datasets[31,32]. `LOL` therefore enables researchers to side-step hotly debated preprocessing issues by hardly preprocessing at all, and instead simply applying `LOL` to the data in its native dimensionality.

## Discussion
We have introduced a very simple methodology to improve performance on supervised learning problems with wide data (that is, big data where dimensionality is at least as large as sample size) by using class-conditional moments to estimate a low rank projection under a generalized framework, `XOX`. In particular, `LOL` uses both the difference of the means and the class-centered covariance matrices, which enables it to outperform `PCA`, as well as existing supervised linear classification schemes, in a wide variety of scenarios without incurring any meaningful additional computational cost. Straightforward generalizations enable robust and nonlinear variants by using robust estimators and/or class specific covariance estimators. Our open source implementation optimally scales to terabyte datasets. Moreover, the intuition can be extended for both hypothesis testing and regression (see Supplementary Note 6 for additional numerical examples in these settings).

Two commonly applied approaches in these settings are `PLS` and `CCA`. `CCA` is equivalent to `rrLDA` whenever *p* < *n*, which is not of interest here. When *p* ≥ *n*, `CCA` and `rrLDA` are not equivalent; however, in such settings, `CCA` exhibits the "maximal

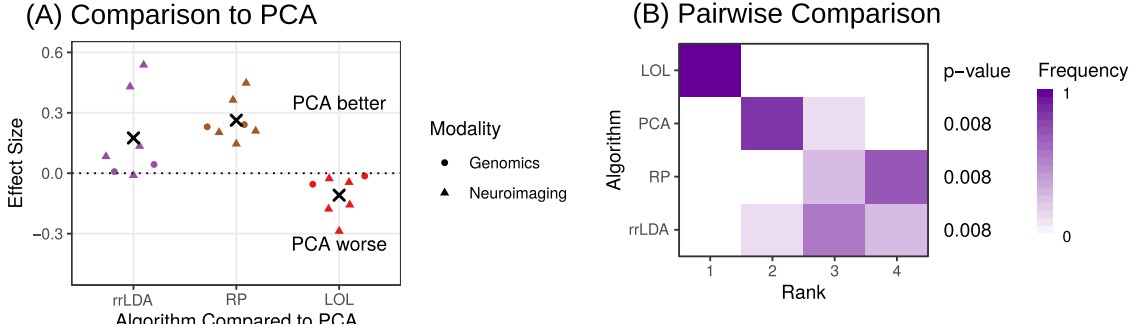

**Fig. 3 Comparing various dimensionality reduction algorithms on two real datasets: neuroimaging and genomics. A** Beeswarm plots show the classification performance of each technique with respect to PCA at the optimal number of embedding dimensions, the number of embedding dimensions with the lowest misclassification rate. Performance is measured by the effect size, defined as $\kappa(\mathrm{LDA} \circ \mathrm{PCA}) - \kappa(\mathrm{LDA} \circ \mathrm{embed})$, where $\kappa$ is Cohen's Kappa, and embed is one of the embedding techniques compared to PCA. Each point indicates the performance of PCA relative the other technique on a single dataset, and the sample size-weighted average effect is indicated by the black "x." LOL always outperforms PCA and all other techniques. **B** Frequency histograms of the relative ranks of each of the embedding techniques on each dataset after classification, where a 1 indicates the best relative classification performance and a 4 indicates the worst relative classification performance, after embedding with the technique indicated. Projecting first with LOL provides a significant improvement over competing strategies (Wilcoxon signed-rank test, $n = 7$, $p$ value $= 0.008$) on all benchmark problems.

data piling problem"[33] (see Supplementary Note 2.6 for details). Specifically, all the points in each class are projected onto the exact same point. This results in severe overfitting of the data, yielding poor empirical performance in essentially all settings we considered here (the first dimension of CCA is typically worse even than the difference of the means). While PLS does not exhibit these problems, it lacks strong theoretical guarantees and simple geometric intuition. In contrast to XOX, neither CCA nor PLS enable straightforward generalizations, such as when there are outliers or the discriminant boundary is quadratic (see Fig. 1). Further, across all simulations, XOX outperforms both of these approaches, sometimes quite dramatically (for example, XOX outperforms CCA on over all of the simulations considered). Finally, no scalable nor parallelized implementations are readily available for these methods (see Fig. 2). One could use stochastic gradient descent with penalties to solve these other optimization problems, but they would still need to tune the penalty parameter which would be quite computationally costly. Neither PLS nor CCA could be successfully run on the massive neuroimaging dataset nor the amplicon-level genomics dataset using readily-available tools.

Many previous investigations have addressed similar challenges. The celebrated Fisherfaces paper was the first to compose Fisher's LDA with PCA (equivalent to PCA in this manuscript)[34]. The authors showed via a sequence of numerical experiments the utility of projecting the data using PCA prior to classifying with LDA. We extend this work by adding a supervised component to the initial projection. Moreover, we provide the geometric intuition for why and when incorporating supervision is advantageous, with numerous examples demonstrating its superiority, and theoretical guarantees formalizing when LOL outperforms PCA. The "sufficient dimensionality reduction" literature has similar insights, but a different construction that typically requires the dimensionality to be smaller than the sample size[35–39] (although see[40] for some promising work). More recently, communication-inspired classification approaches have yielded theoretical bounds on linear and affine classification performance[41]; they do not, however, explicitly compare different projections, and the bounds we provide are more general and tighter. Moreover, none of the above strategies have implementations that scale to millions or billions of features. Recent big data packages are designed for millions or billions of samples[42,43]. In biomedical sciences, however, it is far more common to have

tens or hundreds of samples, and millions or billions of features (e.g., genomics or connectomics).

Most manifold learning methods, while exhibiting both strong theoretical[44–46] and empirical performance, are typically fully unsupervised. Thus, in classification problems, they discover a low-dimensional representation of the data, ignoring the labels. This approach can be highly problematic when the discriminant dimensions and the directions of maximal variance in the learned manifold are not aligned (see Fig. 4 for some examples). Moreover, nonlinear manifold learning techniques tend to learn a mapping from the original samples to a low-dimensional space, but do not learn a projection, meaning that new samples cannot easily be mapped onto the low-dimensional space, a requirement for supervised learning. Deep learning methods[6] can easily be supervised, but they tend to require huge sample sizes, lack theoretical guarantees, or are opaque "black-boxes" that are insufficient for many biomedical applications. This yields a dearth of "out of the box" supervised scalable dimensionality reduction techniques with strong theoretical guarantees with respect to classification performance bounds designed for wide datasets. Random forests circumvent many of these problems, but implementations that operate on millions of dimensions do not exist[47], and often produce embeddings that perform no better than PCA on wide datasets (Fig. 3).

Other approaches formulate an optimization problem, such as projection pursuit[48] and empirical risk minimization[49]. These methods are limited because they are prone to fall into local minima, require costly iterative algorithms, lack any theoretical guarantees on classification accuracy[49]. Feature selection strategies, such as higher criticism thresholding[50] effectively filter the dimensions, possibly prior to performing PCA on the remaining features[51]. These approaches could be combined with LOL in ultrahigh-dimensional problems. Similarly, another recently proposed supervised PCA variant builds on the elegant Hilbert–Schmidt independence criterion[52] to learn an embedding[53]. Our theory demonstrates that under the Gaussian model, composing this linear projection with the difference of the means will improve subsequent performance under general settings, implying that this will be a fertile avenue to pursue. A natural extension to this work would therefore be to estimate a Gaussian mixture model per class, rather than simply a Gaussian per class, and project onto the subspace spanned by the collection of all Gaussians.

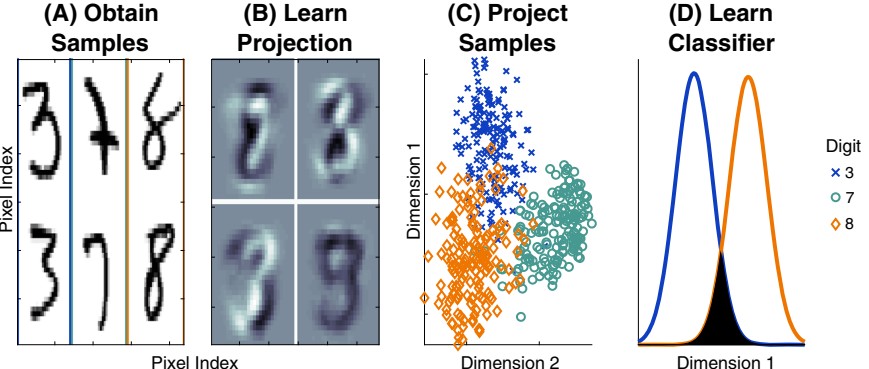

**Fig. 4 Schematic illustrating linear optimal low-rank (LOL) as a supervised manifold learning technique. A** 300 training samples of the numbers 3, 7, and 8 from the MNIST dataset (100 samples per digit); each sample is a $28 \times 28 = 784$ dimensional image (boundary colors are for visualization purposes). **B** The first four projection matrices learned by LOL. Each is a linear combination of the sample images. **C** Projecting 500 new (test) samples into the top two learned dimensions; digits color coded as in (**A**). LOL-projected data from three distinct clusters. **D** Using the low-dimensional data to learn a classifier. The estimated distributions for 3 and 8 of the test samples (after projecting data into two dimensions and using LDA to classify) demonstrate that 3 and 8 are easily separable by linear methods after LOL projections (the color of the line indicates the digit). The filled area is the estimated error rate; the goal of any classification algorithm is to minimize that area. LOL is performing well on this high-dimensional real data example.

In conclusion, the key XOX idea, appending class-conditional moment estimates to convert unsupervised manifold learning to supervised manifold learning, has many potential applications and extensions. We have presented the first few, including LOL, QOQ, and RLOL, which demonstrated the flexibility of XOX under both theoretical and benchmark settings. Incorporating additional nonlinearities via higher order moments, kernel methods[54], ensemble methods[55] such as random forests[56], and multiscale methods[46] are all of immediate interest.

## Methods

**Supervised manifold learning**. A general strategy for supervised manifold learning is schematized in Fig. 4, and outlined here. Step (**A**): Obtain or select $n$ training samples of high-dimensional data. For concreteness, we use one of the most popular benchmark datasets, the MNIST dataset[57]. This dataset consists of images of hand-written digits 0 through 9. Each image is represented by a $28 \times 28$ matrix, which means that the observed dimensionality of the data is $p = 28^2 = 784$. Because we are motivated by the $n \ll p$ scenario, we subsample the data to select $n = 300$ examples of the numbers 3, 7, and 8 (100 of each). Step (**B**): Learn a "projection" that maps the high-dimensional data to a low-dimension representation. One can do so in a way that ignores which images correspond to which digit (the "class labels"), as PCA and most manifold learning techniques do, or try to use the labels, as LDA and sparse methods do. LOL is a supervised linear manifold learning technique that uses the class labels to learn projections that are linear combinations of the original data samples. Step (**C**): Use the learned projections to map high-dimensional data into the learned lower-dimensional space. This step requires having learned a projection that can be applied to new (test) data samples for which we do not know the true class labels. Nonlinear manifold learning methods typically cannot be applied in this way (though see[58]). LOL, however, can project new samples in such a way as to separate the data into classes. Step (**D**): Using the low-dimensional representation of the data, learn a classifier. A good classifier correctly identifies as many points as possible with the correct label. For these data, when LDA is used on the low-dimensional data learned by LOL, the data points are mostly linearly separable, yielding a highly accurate classifier.

**The geometric intuition of LOL**. To build intuition for situations when LOL performs well, and when it does not, we consider the simplest high-dimensional classification setting. We observe $n$ samples $(x_i, y_i)$, where $x_i$ are $p$ dimensional feature vectors, and $y_i$ is the binary class label, that is, $y_i$ is either 0 or 1. We assume that both classes are distributed according to a multivariate Gaussian distribution, the two classes have the same identity covariance matrix (all features are uncorrelated with unity variance), and data from either class is equally likely, so that the only difference between the classes is their means. In this scenario, the optimal low-dimensional projection is analytically available: it is the dot product of the difference of means and the inverse covariance matrix, commonly referred to as Fisher's Linear Discriminant Analysis (LDA)[59] (see Supplementary Note 1.2 for derivation). When the distribution of the data is unavailable, as in all real data problems, machine learning methods can be used to estimate the parameters. Unfortunately, when $n < p$, the estimated covariance matrix will not be invertible (because the solution to the underlying mathematical problem is under specified),

so some other approach is required. As mentioned above, PCA is commonly used to learn a low-dimensional representation. PCA uses the pooled sample mean and the pooled sample covariance matrix. The PCA projection is composed of the top $d$ eigenvectors of the pooled sample covariance matrix, after subtracting the pooled mean (thereby completely ignoring the class labels).

In contrast, LOL uses the class-conditional means and class-centered covariance. This approach is motivated by Fisher's LDA, which uses the same two terms, and should therefore improve performance over PCA. More specifically, for a two-class problem, LOL is constructed as follows:

1. Compute the sample mean of each class.
2. Estimate the difference between means.
3. Compute the class-centered covariance matrix, that is, compute the covariance matrix after subtracting the class mean from each point.
4. Compute the eigenvectors of this class-conditionally centered covariance.
5. Concatenate the difference of the means with the top $d - 1$ eigenvectors of class-centered covariance.

Note that the sample class-centered covariance matrix estimates the population covariance, whereas the sample pooled covariance matrix is distorted by the difference of the class means. Further, as discussed in Methods, the class-centered covariance matrix is equivalent to "Reduced Rank LDA"[60] (rrLDA hereafter, which is simply LDA but truncating the covariance matrix). For the theoretical background on LDA and rrLDA, a formal definition of LOL, and detailed description of the simulation settings that follow, see Supplementary Notes 1, 2, and 3, respectively. Figure 5 shows three different examples of 100 data points sampled from a 1000 dimensional Gaussian to geometrically illustrate the intuition that motivated LOL. In each case, all dimensions are uncorrelated with one another, and all classes are equally likely with the same covariance; the only difference between the classes are their means.

Figure 5a shows "stacked cigars", in which the difference between the means and the direction of maximum variance are large and aligned with one another. This is an idealized setting for PCA, because PCA finds the direction of maximal variance, which happens to correspond to the direction of maximal separation of the classes. rrLDA performs well here too, for the same reason that PCA does. Because all dimensions are uncorrelated, and one dimension contains most of the information discriminating between the two classes, this is also an ideal scenario for sparse methods. Indeed, ROAD, a sparse method designed for precisely this scenario, does an excellent job finding the most useful dimensions[12]. LOL, using both the difference of means and the directions of maximal variance, also does well. To calibrate all of these methods, we also show the performance of the optimal classifier.

Figure 5b shows an example that is worse for PCA. In particular, the variance is getting larger for subsequent dimensions, while the magnitude of the difference between the means is decreasing with dimension. Because PCA operates on the pooled sample covariance matrix, the dimensions with the maximum difference are included in the estimate, and therefore, PCA finds some of them, while also finding some of the dimensions of maximum variance. The result is that PCA performs fairly well in this setting. rrLDA, however, by virtue of subtracting out the difference of the means, is now completely at chance performance. ROAD is not hampered by this problem; it is also able to find the directions of maximal discrimination, rather than those of maximal variance. Again, LOL, by using both the means and the covariance, does extremely well.

Figure 5c is exactly the same as Fig. 5b, except the data have been randomly rotated in all 1000 dimensions. This means that none of the original features have

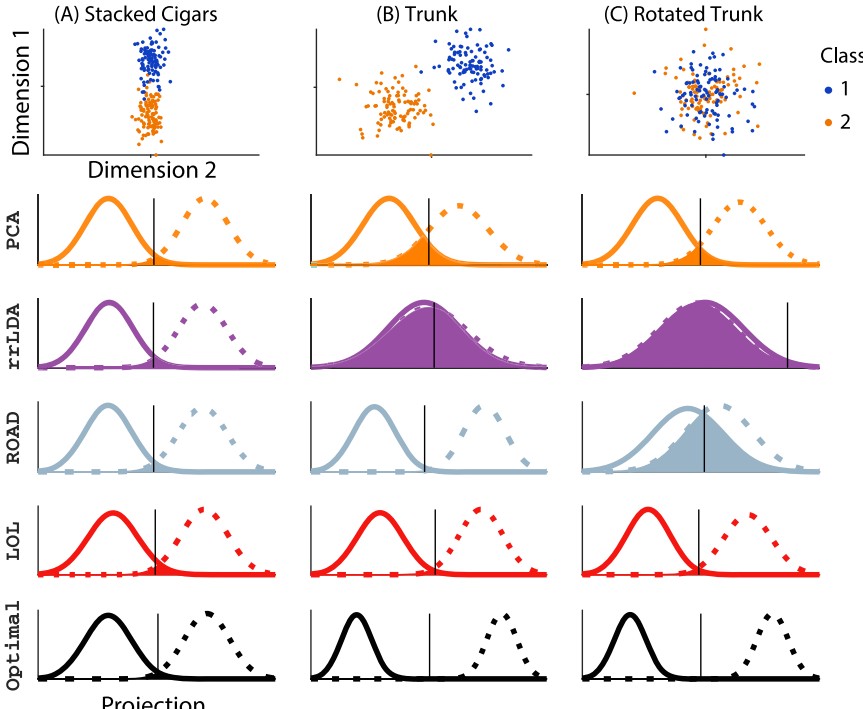

**Fig. 5 LOL achieves near-optimal performance for three different multivariate Gaussian distributions, each with 100 samples in 1000 dimensions.** For each approach, we project into the top three dimensions, and then use LDA to classify 10,000 new samples. The six rows show (from top to bottom): Row 1: A scatter plot of the first two dimensions of the sampled points, with class 0 and 1 as orange and blue dots, respectively. The next rows each show the estimated posterior for class 0 and class 1, in solid and dashed lines, respectively. The overlap of the distributions---which quantifies the magnitude of the error---is filled. The black vertical line shows the estimated threshold for each method. The techniques include: PCA; reduced rank LDA(rrLDA), a method that projects onto the top $d$ eigenvectors of sample class-conditional covariance; ROAD, a sparse method designed specifically for this model; LOL, our proposed method; and the Bayes optimal classifier. **A** Stacked Cigars The mean difference vector is aligned with the direction of maximal variance, and is mostly concentrated in a single dimension, making it ideal for PCA, rrLDA, and sparse methods. In this setting, the results are similar for all methods, and essentially optimal. **B** Trunk The mean difference vector is orthogonal to the direction of maximal variance; PCA performs worse and rrLDA is at chance, but sparse methods and LOL can still recover the correct dimensions, achieving nearly optimal performance. **C** Rotated Trunk Same as (**B**), but the data are rotated; in this case, only LOL performs well. Note that LOL is closest to Bayes optimal in all three settings.

much information, but rather, linear combinations of them do. This is evidenced by observing the scatter plot, which shows that the first two dimensions fail to disambiguate the two classes. PCA performs even worse in this scenario than in the previous one. rrLDA is rotationally invariant (see Supplementary Note 2.4 for details), so still performs at chance levels. Because there is no small number of features that separate the data well, ROAD fails. LOL performs as well here as it does in the other examples.

**When is LOL better than PCA and other supervised linear methods?** We desire theoretical confirmation of the above numerical results. To do so, we investigate when LOL is "better" than other linear dimensionality reduction techniques. In the context of supervised dimensionality reduction or manifold learning, the goal is to obtain low dimensional representation that maximally separates the two classes, making subsequent classification easier. Chernoff information quantifies the dissimilarity between two distributions. Therefore, we can compute the Chernoff information between distribution of the two classes after embedding to evaluate the quality of a given embedding strategy. As it turns out, Chernoff information is the exponential convergence rate for the Bayes error[61], and therefore, the tightest possible theoretical bound. The use of Chernoff information to theoretically evaluate the performance of an embedding strategy is novel, to our knowledge, and leads to the following main result:

**Main theoretical result**. LOL is always better than or equal to rrLDA under the Gaussian model when $p \geq n$, and better than or equal to PCA (and many other linear projection methods) with additional (relatively weak) conditions. This is true for all possible observed dimensionalities of the data, and the number of dimensions into which we project, for sufficiently large sample sizes. Moreover, under relatively weak assumptions, these conditions almost certainly hold as the number of dimensions increases.

Formal statements of the theorems and proofs required to substantiate the above result are provided in Methods. The condition for LOL to be better than PCA is essentially that the $d^{th}$ eigenvector of the pooled sample covariance matrix has less information about classification than the difference of the means vector. The

implication of the above theorem is that it is better to incorporate the mean difference vector into the projection matrix, rather than ignoring it, under basically the same assumptions that motivate PCA. The degree of improvement is a function of the dimensionality of the feature set $p$, the number of samples $n$, the projection dimension $d$, and the parameters, but the existence of an improvement—or at least no worse performance—is independent of those factors.

## Data availability

Data used within this manuscript are available from https://neurodata.io/lol/and https://neurodata.io//mri.

## Code availability

MATLAB, R, and Python code for the experiments performed in this manuscript and a docker container for FlashLOL are available from https://neurodata.io/lol/, and an R package is available on the Comprehensive R Archive Network (CRAN)[62].

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

## Acknowledgements

The authors are grateful for the support by the XDATA program of the Defense Advanced Research Projects Agency (DARPA) administered through Air Force Research Laboratory contract FA8750-12-2-0303; DARPA GRAPHS contract N66001-14-1-4028;

and DARPA SIMPLEX program through SPAWAR contract N66001-15-C-4041 and DARPA Lifelong Learning Machines program through contract FA8650-18-2-7834.

## Author contributions

M.T. and M.M. contributed theoretical results, D.Z. and R.B. devised the semi-external memory implementation, C.D. procured relevant genomics datasets, J.T.V. and E.W.B. wrote the paper, E.W.B. developed the experiments and R package, J.T.V. supervised.

## Competing interests

The authors declare no competing interests.
