## [Peer Review File · Nature Communications]

REVIEWER COMMENTS

Reviewer #1 (Remarks to the Author):

This paper proposes a supervised dimension reduction approach called 'linear optimal low-rank (LOL)' projection that combines both class-conditional means and principal component directions to construct projection matrices. Theoretically, the authors show that the LOL projection preserves more Chernoff information than those by PCA and LDA, implying higher discriminative capabilities. Simulation study illustrates the superiority of LOL (or its variants) over PCA, LDA and ROAD in terms of misclassification errors under various embedding dimensions. Real data analysis on high-dimensional brain imaging and genomic datasets lends further support to the statistical superiority and computational efficiency of LOL.

This work studies a very important problem of modern data challenges. The key idea of the proposed algorithm is to incorporate class-conditional means into dimension reduction. This paper has solid theory and sufficient numerical study. Below I list my comments on the paper:

1. From my perspective, the major limit of LOL is that it incorporates only class-conditional means to the dimension reduction. What if the classes share (nearly) the same means? What will be the consequences if one incorporates other class-specific statistics, such as elementwise medians, into the dimension reduction procedure? I understand that incorporating the mean structure is a natural starting point and easy to analyze under Gaussian setups, but I expect more justification of this choice, particularly in the real data contexts.
2. In Figure 3, it is interesting to compare all the methods under different embedding dimensions d , but I am more interested in comparison between all the methods with d tuned by cross validation as in the real data analysis. I believe the misclassification error with tuned d is a more convincing criteria to look at.
3. Finally, in Figure 3, I see two variants of LOL called QOQ and RLOL that handle heterogeneous covariance matrices across classes and outliers respectively. They look interesting, and I suggest the authors elucidate these methods more. Particularly, QOQ also incorporates class-conditional covariance to construct projection matrices. This makes me wonder whether the authors can unify LOL, QOQ and RLOL in a general framework that uses class-conditional statistics to perform dimension reduction.

Typos:

Figure 3's caption: 'The left column ... The right column ...' should be changed to 'The first row ... The second row ...'

Reviewer #2 (Remarks to the Author):

Summary of paper

This paper introduces a novel, supervised, linear dimensionality reduction method coined LOL (Linear optimal low-rank projection). LOL extends the popular, unsupervised, PCA method by (a) utilizing the class-conditional covariance and (b) appending class-conditional mean differences as features in the low-rank projection. The paper empirically demonstrates

the effectiveness of LOL for classification using LDA compared to PCA and random projections. The paper further develops a finite sample difference between LOL+LDA and PCA+LDA, as well as the difference in Chernoff information captured between each method for 2-class classification.

Summary of review

I believe that this paper should be published in this venue; however, I would prefer to see some edits made to the paper before publication. I find the paper to be novel and insightful. I believe the LOL dimensionality reduction method is useful for improving 2-class classification performance in a low-sample, high-dimensional setting; while maintaining sound theoretical improvement guarantees over PCA. I am unconvinced of the applicability of LOL beyond the 2-class classification setting and I would have preferred to see a deeper investigation of other dimensionality reduction methods.

Theorem 1 suggests that the difference in Chernoff information between LOL and PCA is inversely proportional to the size of the d 'th eigenvalue, implying (as stated in the main body of the paper) that the degree of benefit afforded by LOL is based on how much information is gained due to replacing the eigenvector of smallest explained variance with the mean difference vector. In general for C -class classification, this bound will become inversely proportional to the $[d, d - 1, \dots, d - (C - 2)]$ eigenvalues. Can the theory be extended to include the general C -class setting? How restrictive do the conditions for LOL improving over PCA get as C grows larger? In the 2-class setting, Theorem 1 appears to state that LOL always improves over PCA because the difference in Chernoff information is always greater than zero. Does this hold for $C > 2$?

The empirical section only investigates 2-class classification except for Figure 3a, which demonstrates that PCA outperforms LOL for a high number of embedded dimensions (which I'll call d). As $C \gg 2$, then how does this impact the value of d at which PCA improves over LOL? My presumption is that this value of d will decrease as C increases because LOL replaces the bottom eigenvectors with $C-1$ mean difference vectors. I would like to see this further investigated to know whether LOL is robust to classification settings with larger values of C (either empirically or by extending the theory).

This paper also discounts many dimensionality reduction methods and investigates only PCA and random projection methods. While I agree with the paper that many dimensionality reduction methods do not meet the goals stated by this paper, I do believe that some further exploration would be useful to more appropriately place this paper within the literature. For instance, I am unconvinced that supervised autoencoders should be entirely discounted. While I agree that deep non-linear SAEs are uninterpretable black-box methods with too many tunable parameters, I wonder how a fully linear SAE with only a single hidden layer would perform compared to PCA/LOL. In fact, such an AE (without the supervised component) has been shown to be the same function class as PCA, so how does the corresponding SAE compare to LOL? I would argue that the interpretability of this SAE is no worse (or better) than the interpretability of LOL.

Similar to the discussion on supervised autoencoders, I would have liked to see further discussion on supervised dictionary learning (the seminal SDL paper being cited by this paper, but not discussed). Extending SDL to out-of-sample prediction has been widely investigated and the goal of SDL is to directly parameterize the "projection" matrix that this paper seeks to hand-design. While the theory in this paper suggests that the hand-designed projection matrix is better than PCA, I wonder if it is likewise better than a direct

parameterization/optimization using SDL (note this relationship can likewise be shown with SAEs, though it is obscured).

Paper describing SAEs for out-of-sample supervised classification (for transparency, note I am an author on this paper):

Le, Lei, Andrew Patterson, and Martha White. "Supervised autoencoders: Improving generalization performance with unsupervised regularizers." *Advances in Neural Information Processing Systems*. 2018.

Paper describing SDL for out-of-sample supervised classification:

Mairal, Julien, Francis Bach, and Jean Ponce. "Task-driven dictionary learning." *IEEE transactions on pattern analysis and machine intelligence* 34.4 (2011): 791-804.

Minor comments/question

LOL can be seen as making two modifications to PCA. It uses the class-conditional covariance matrix to choose the d highest variance eigenvectors, then it also replaces $C-1$ of those d eigenvectors with the difference between class conditional means. How do either of these modifications perform in isolation? Based on my understanding of Theorem 1, using the class conditional means is the most important novelty, but does that hold in practice as well?

In Figure 3, why does only LOL have a "robust" version? The performance gain of LOL in Figure 3b is negligible and much worse than many competitors, but "robust" LOL does quite well. Would using a robust PCA have the same effect? Or can only LOL be made robust in this way? The robust LOL method is never described that I could find it. My assumption is that you replaced the class-conditional mean with a class-conditional median. Likewise in Figure 3c, why not compare QOQ with QDA as well as QDA+PCA? Would this significantly outperform QOQ? Given that LDA and PCA both outperform LOL, I would expect that the quadratic versions would likewise outperform the quadratic version of LOL. If this is true, then that means LOL only outperformed its baseline in one of three simulated settings and only for small values of d in that one setting.

In Figure 4, why can't PCA use the randomized approximate algorithm for eigendecomposition? Would PCA not also be able to benefit greatly in terms of computational performance, especially considering that LOL is simply PCA with a class-conditional mean centering preprocessing step (e.g. LOL and PCA rely on the exact same eigendecomposition code, LOL just has a subtraction step first).

What is Figure 5 showing for the LDA columns? Since RR-LDA is equivalent to LDA composed with PCA, then it can't be showing the reduced rank form. I'm not sure that LDA can be composed with LDA, that doesn't make sense to me (thus the label is wrong, I believe), so what is the LDA column in Figure 5a and row in Figure 5b? Is it LDA applied to the raw dataset without dimensionality reduction?

In Figure 6, you show LOL used for a regression problem. This is quite interesting, and in general I would love to see more exploration of LOL used with other classification algorithms (other than LDA). However, how can LOL be used for regression? How do you compute the class-conditional covariance and means for a regression problem? Without these, I would expect LOL to revert back to PCA.

I believe appendix B has a few typos:

- B.II: g^d_{LDA} is defined twice. I believe the first definition should be labelled g_{LDA}
- B.III: δ is defined as a matrix of size $C-2 * p$ instead of $C-1 * p$. Should the last index in this array be δ_C instead of δ_{C-1} ?
- B.III: A^D_{LOL} is defined twice and in two different ways. The first definition assumes two-class and the second definition is more general for any number of C .

Andrew Patterson

Reviewer #3 (Remarks to the Author):

Summary:

The paper discusses the use of linear projection (PCA) per class followed by LDA to preserve the discriminatory power of features. The LOL method ensures that the projected data is useful for the subsequent linear classification problem.

Comments:

Unfortunately, the idea of appending the mean difference vector to features and class-centering to convert unsupervised manifold learning to a supervised manifold learning setting are not a substantial enough contribution to ML methodology. The claims of scalability are not interesting either. The draft has not been proofread properly has as elementary mistakes even in the background/preliminary sections such as A.I and A.II.

>> Existing linear and nonlinear dimensionality reduction methods either are not supervised, scale poorly to operate in big data regimes, lack theoretical guarantees, or are “black-box” methods unsuitable for many applications.

This is an exaggerated mis-characterization of work in this area. There are multiple papers that address scalability, address theory of such dimensionality reduction, as well as are quite interpretable and transparent.

>> LOL achieves achieves state-of-the-art classification accuracy

This seems to be an exaggerated claim. Partly because restricting oneself to linear classification settings already limits the model representation power. What does state of the art even mean here?

Additional Comments:

- The first expression for $g_{\text{LDA}}(x)$ in pg 19 looks wrong. The first term should also be a function of y .
- Why did the narrative change to two class instead of the general setting in A.II pg 19 even before calling out for the two class setting (i.e., why $\pi_0 = \pi_1$ before talking about the two-class setting)?
- Lemma 1 is trivial/not new.

- Eq (2) pg 20 is not new.

- What is the difference between L_A^d and L_A ? Please be consistent in notation.

- In Eq (3), is Σ and δ (which determine Σ_A and δ_A) already known? This should have been clarified as this is very importantly and makes the optimization problem difficult/easy.

- >> because we do not know how to evaluate the integral analytically

Why is this a deal breaker. There are many problems where the population optimal estimation or model is not computable? Also, can you comment on what type of problem this is? non-convex? for what additional assumptions could it become convex?

- What is \mathcal{A}^d and \mathcal{A} ? If they are the same, then how can $\delta^T \Sigma^{-1}$ belong to this set. The dimensions don't match, so Lemma 2 is incorrect.

- Lemma 2 proof: what is δ_B ?

The point-by-point is organized as follows. Comments from reviewers are enumerated in black text. Our responses are provided in blue text. Relevant text changes which specifically address the reviewer's comment are indicated by the red text within the screenshot (and the corresponding manuscript).

Reviewer #1 (Remarks to the Author):

This paper proposes a supervised dimension reduction approach called 'linear optimal low-rank (LOL)' projection that combines both class-conditional means and principal component directions to construct projection matrices. Theoretically, the authors show that the LOL projection preserves more Chernoff information than those by PCA and LDA, implying higher discriminative capabilities. Simulation study illustrates the superiority of LOL (or its variants) over PCA, LDA and ROAD in terms of misclassification errors under various embedding dimensions. Real data analysis on high-dimensional brain imaging and genomic datasets lends further support to the statistical superiority and computational efficiency of LOL.

This work studies a very important problem of modern data challenges. The key idea of the proposed algorithm is to incorporate class-conditional means into dimension reduction. This paper has solid theory and sufficient numerical study. Below I list my comments on the paper:

1. From my perspective, the major limit of LOL is that it incorporates only class-conditional means to the dimension reduction. What if the classes share (nearly) the same means? What will be the consequences if one incorporates other class-specific statistics, such as elementwise medians, into the dimension reduction procedure? I understand that incorporating the mean structure is a natural starting point and easy to analyze under Gaussian setups, but I expect more justification of this choice, particularly in the real data contexts.

Thank you for your feedback. We have updated our wording in the abstract, introduction, the content discussion of Figure 3(C), and discussion to better emphasize our contribution as XOX, a procedure for incorporating (in the general case) class conditional moment estimates into a supervised embedding, of which LOL, QOQ, and Robust LOL (RLOL) are special cases. Our goal was simply to use LOL as an example of such a framework, due to the fact that we can make strong theoretical guarantees while maintaining strong empirical performance despite the triviality of the modification. If the classes share nearly the same means, variants of XOX incorporating different class-conditional moments may be sensible, such as QOQ, as highlighted in Figure 3(C). XOX could incorporate any class-conditional moment estimates, such as the class-conditional medians as you mention, we simply chose not to for illustration.

To solve key biomedical problems, experimentalists now routinely measure millions or billions of features (dimensions) per sample, with the hope that data science techniques will be able to build accurate data-driven inferences. Because sample sizes are typically orders of magnitude smaller than the dimensionality of these data, valid inferences require finding a low-dimensional representation that preserves the discriminating information (e.g., whether the individual suffers from a particular disease). There is a lack of interpretable supervised dimensionality reduction methods that scale to millions of dimensions with strong statistical theoretical guarantees. **We introduce an approach, XOX**, to extending principal components analysis by incorporating class-conditional moment estimates into the low-dimensional projection. The simplest version, “Linear Optimal Low-rank” projection (LOL), incorporates the class-conditional means. We prove, and substantiate with both synthetic and real data benchmarks, that LOL and its generalizations **in the XOX framework** lead to improved data representations for subsequent classification, while maintaining computational efficiency and scalability. Using multiple brain imaging datasets consisting of >150 million features, and several genomics datasets with >500,000 features, LOL achieves outperforms other scalable algorithms in terms of accuracy, while only requiring a few minutes on a standard desktop computer.

To address these issues, **we developed a technique for incorporating class-conditional moment estimates, XOX, the simplest example of which is LOL**. *The key intuition behind LOL is that we can jointly use the means and variances from each class (like LDA and CCA), but without requiring more dimensions than samples (like PCA), or restrictive sparsity assumptions.* Using random matrix theory, we are able to prove that when the data are sampled from a Gaussian, LOL finds a better low-dimensional representation than PCA, LDA, CCA, and other linear methods. Under relatively relaxed assumptions, this is true regardless of the dimensionality of the features, the number of samples, or the number of dimensions in which we project. We then demonstrate the superiority of techniques derived using the XOX approach—including (i) LOL, (ii) **a variant of XOX which allows greater flexibility of the class-conditional covariances called QOQ**, and (iii) **a robust variant of LOL called RLOL**—over other methods numerically on a variety of simulated settings including several not following the theoretical assumptions. Finally, we show that on several 500 gigabyte neuroimaging datasets, and several multi-gigabyte genomics datasets, LOL achieves superior accuracy at lower dimensions while requiring only a few minutes of time on a single workstation.

ances. Another variant of LOL, Quadratic Optimal QDA (QOQ), computes the eigenvectors separately for each class, concatenates them (sorting them according to their singular values), and then classifies with QDA instead of LOL. For all three scenarios, either LOL—or its extended variants RLOL and QOQ—achieves a misclassification rate comparable to or lower than other methods, for all dimensions. **These three results demonstrate how straightforward generalizations of LOL under the XOX framework which incorporate alternate or robust moment estimates** can dramatically improve performance over other projection methods. This is in marked contrast to other approaches, for which such flexibility is either not available, or otherwise problematic.

We have introduced a very simple methodology to improve performance on supervised learning problems with wide data (that is, big data where dimensionality is at least as large as sample size) by using class-conditional moments to estimate a low rank projection under a generalized framework, `XOX`. In particular, `LOL` uses both the difference of the means and the class-centered covariance matrices, which enables it to outperform `PCA`, as well as existing supervised linear classification schemes, in a wide variety of scenarios without incurring any meaningful additional computational cost. Straightforward generalizations enable robust and nonlinear variants by using robust estimators and/or class specific covariance estimators. Our open source implementation optimally scales to terabyte datasets. Moreover, the intuition can be extended for both hypothesis testing and regression (see Appendix F for additional numerical examples in these settings).

For `RLOL`, we *did* use the class medians as the “robust estimate of the first moment” (as well as a robust estimate of the second moment); we have better clarified in the text.

Figure 3 shows a two-dimensional scatterplot (left) and misclassification rate versus dimensionality (right) for each simulation. The top $C - 1$ embedding dimensions for `LOL` correspond to the performance after projection onto the class-conditional means, and `rrLDA` corresponds to the performance of projection onto the class-conditional covariance matrix. Figure 3A shows a three class generalization of the Trunk example from Figure 2B. `LOL` can trivially be extended to more than two classes (see Section B for details), unlike `ROAD` which only operates in a two-class setting. Figure 3B shows a two-class example with many outliers, as is typical in modern biomedical datasets. A variant of `LOL`, “Robust `LOL`” (`RLOL`), replaces the standard estimators of the mean and covariance with robust variants (the class-conditional medians and robust covariance [27] respectively), thereby dramatically improving performance over `LOL` (and other techniques) in noisy settings. Hereafter, `LOL` will refer to the version of `LOL` with a robust estimate of the first moment, and a truncated estimate of the second moment, as a robust first moment tends to make little difference when a robust estimate was not necessary, and improved performance when a robust estimate was warranted. We do not use a robust estimate of

the second moment, as typical robust estimates of the second moment available in standard numerical packages require $d < n$, which is unsuitable for wide data. Figure 3C shows an example that does

Upon further review of all experiments, moreover, we realized that while a robust estimate of the second moment limits us to be able to only look at problems where the dimensionality is less than or equal to the sample size in standard available numerical packages (R::robust), a robust estimate of the first moment has no such requirement, so we have replaced all instances of the `LOL` simulations with a variant featuring a robust estimate of the first moment and a truncated estimate of the second moment via truncated svd. This had no impact on any results contained herein other than the Robust simulation, so we believe it is a simpler story to tell than to have separate embedding procedures for what is otherwise a linear classification problem in that particular figure.

Figure 3(C) shows a benchmark simulation in which the simulations share the same means (but have orthogonal covariance).

2. In Figure 3, it is interesting to compare all the methods under different embedding dimensions d , but I am more interested in comparison between all the methods with d tuned by cross

validation as in the real data analysis. I believe the misclassification error with tuned d is a more convincing criteria to look at.

For each simulation, we have added a point corresponding to the optimal number of embedding dimensions, with the number of embedding dimensions tuned as-in the real data analysis. The misclassification error is seen to be approximately the lowest using LOL or one of its variants in all three settings.

3. Finally, in Figure 3, I see two variants of LOL called QOQ and RLOL that handle heterogeneous covariance matrices across classes and outliers respectively. They look interesting, and I suggest the authors elucidate these methods more. Particularly, QOQ also incorporates class-conditional covariance to construct projection matrices. This makes me wonder whether the authors can unify LOL, QOQ and RLOL in a general framework that uses class-conditional statistics to perform dimension reduction.

Thank you for this extremely insightful feedback. As mentioned in point 1, we attempted to unify the approach of “the use of class-conditional moments for embedding” into a unified framework, XOX, which allows one to fairly seamlessly generalize LOL to situations in which other class-conditional moments may be of use. We believe this readily harmonizes LOL, QOQ, RLOL, and other approaches discussed within the manuscript under a single, harmonious, framework.

Typos:

Figure 3's caption: “The left column ... The right column ...” should be changed to “The first row ... The second row ...”

We appreciate this feedback and have updated the caption accordingly.

Reviewer #2 (Remarks to the Author):

Summary of paper

This paper introduces a novel, supervised, linear dimensionality reduction method coined LOL (Linear optimal low-rank projection). LOL extends the popular, unsupervised, PCA method by (a) utilizing the class-conditional covariance and (b) appending class-conditional mean differences as features in the low-rank projection. The paper empirically demonstrates the effectiveness of LOL for classification using LDA compared to PCA and random projections. The paper further develops a finite sample difference between LOL+LDA and PCA+LDA, as well as the difference in Chernoff information captured between each method for 2-class classification.

Summary of review

I believe that this paper should be published in this venue; however, I would prefer to see some edits made to the paper before publication. I find the paper to be novel and insightful. I believe the LOL dimensionality reduction method is useful for improving 2-class classification performance in a low-sample, high-dimensional setting; while maintaining sound theoretical improvement guarantees over PCA. I am unconvinced of the applicability of LOL beyond the 2-class classification setting and I would have preferred to see a deeper investigation of other dimensionality reduction methods.

Theorem 1 suggests that the difference in Chernoff information between LOL and PCA is inversely proportional to the size of the d 'th eigenvalue, implying (as stated in the main body of the paper) that the degree of benefit afforded by LOL is based on how much information is gained due to replacing the eigenvector of smallest explained variance with the mean difference vector. In general for C -class classification, this bound will become inversely proportional to the $[d, d - 1, \dots, d - (C - 2)]$ eigenvalues. Can the theory be extended to include the general C -class setting? How restrictive do the conditions for LOL improving over PCA get as C grows larger? In the 2-class setting, Theorem 1 appears to state that LOL always improves over PCA because the difference in Chernoff information is always greater than zero. Does this hold for $C > 2$?

We appreciate this comment. In our attempt to extend a comparison of LOL with PCA and rrLDA under the $C > 2$ setting, we find that whereas PCA and rrLDA only depend on the covariance matrix, LOL also depends on the subspace spanned by the difference of the means. Therefore, a theoretical comparison of embeddings becomes far more complicated, which is a problem we think may warrant further exploration in a theoretical paper of its own. We have added a remark which frames the $C > 2$ class problem to serve as a stepping stone for future work.

Remark 3. The previous comparisons are done for the case of $C = 2$ classes. Extending these comparisons to the case of $C > 2$ classes is, however, non-trivial. More precisely, suppose we have $Y \in \{1, 2, \dots, C\}$ and that, conditional on $Y = c$, $X \sim \mathcal{N}(\mu_c, \Sigma)$ is multivariate normal with mean μ_c and common covariance matrix Σ . Then, given $X = x$, the Bayes optimal classifier for Y is still

$$g_{\text{LDA}}(x) = \underset{y \in \{1, 2, \dots, C\}}{\operatorname{argmin}} \left[\frac{1}{2}(x - \mu_y)^\top \Sigma^{-1}(x - \mu_y) - \log \pi_y \right] = \underset{y \in \{1, 2, \dots, C\}}{\operatorname{argmin}} \left[-x^\top \Sigma^{-1} \mu_y + \frac{1}{2} \mu_y^\top \Sigma^{-1} \mu_y - \log \pi_y \right]$$

Taking $\frac{1}{2} \mu_y^\top \Sigma^{-1} \mu_y - \log \pi_y$ as either a given constant or as an intercept term to be learned or estimated, the reduced-rank LDA for $C > 2$ classes still corresponds to looking at the top d eigenvectors of Σ . That is to say, we transform the predictor variables via $x \mapsto U_d x$ followed by performing LDA on the transformed data. Similarly, the PCA transformation corresponds to using the top d eigenvectors of the pooled covariance matrix $\tilde{\Sigma} = \mathbb{E}[(X - \sum_c \pi_c \mu_c)(X - \sum_c \pi_c \mu_c)^\top]$ followed by performing LDA. Suppose we now compare LOL, rrLDA, and PCA in this multi-class setting. Let $A: X \mapsto AX$ be a linear transformation. Then by Eq. (11) and Eq. (12), the Chernoff information for the transformed data in this multi-class setting is

$$\min_{c \neq c'} \frac{1}{8} (\mu_c - \mu_{c'})^\top A^\top (A \Sigma A^\top)^{-1} A (\mu_c - \mu_{c'}).$$

We now see that, in the case of rrLDA and PCA the linear transformation A depends only on the covariance matrix Σ and $\tilde{\Sigma}$, respectively. That is to say, the linear transformation A does not depend on the choice of c and c' . In contrast, currently for LOL the linear transformation A depends on both Σ as well as $\mu_c - \mu_{c'}$. In other words, there is no single choice for A but rather that A changes as c, c' changes. Direct comparison, in the multi-classes setting, between LOL and either of rrLDA or PCA is thus an open problem that we leave for future work. Finally we note that if we allow the linear transformation for LOL to vary with the classes c and c' , i.e., taking a one-vs-one approach to multi-classes classification, then the results presented in this paper are valid for all pairs c, c' .

The empirical section only investigates 2-class classification except for Figure 3a, which demonstrates that PCA outperforms LOL for a high number of embedded dimensions (which I'll call d). As $C \gg 2$, then how does this impact the value of d at which PCA improves over LOL? My presumption is that this value of d will decrease as C increases because LOL replaces the bottom eigenvectors with $C-1$ mean difference vectors. I would like to see this further investigated to know whether LOL is robust to classification settings with larger values of C (either empirically or by extending the theory).

Thank you for your point; we have added Figure 6 to the supplement to explore the impact on the dimensionality as the number of embedding dimensions is increased, as the number of classes increases. We believe this supports that this figure indicates that LOL is robust to larger values of C under the given empirical setup.

F.I Large numbers of classes

Here, we explore an experiment in which the number of classes increases for a given simulation. We look at the multiclass hump- K problem, described in Section C. In this simulation, while the space spanned by the differences of means conveys more information than the directions of maximal variance, we expect that the shift in the means for a given class at a given dimension should also increase the variance fractionally in that direction as well. Figure 6A shows the simulation setup, for $K = 10$. Figure 6B indicates the misclassification rate as a function of Cohen's Kappa. We use Cohen's Kappa instead of the misclassification rate for direct evaluation since K varies widely across these simulations at a fixed number of total samples $n = 128$, making the difficulty of the problem as K increases two-fold: not only are there more classes, but there are also fewer examples of each class per simulation setting. In all cases, the best random classifier would be the classifier that continually guesses a single class continuously, which has expected accuracy of $\frac{1}{K}$. On all simulations, we see that both PLS and LOL rapidly approach a higher Kappa statistic (better performance relative the random classifier) as they learn the space spanned by the differences of means. PLS rapidly declines in performance as successive dimensions are added, and LOL sees a small performance decline, as successive dimensions should convey no information regarding the class. PCA is able to ultimately identify the space spanned by the differences of the means, but takes far more embedding dimensions to do so, and yields a lower Kappa statistic than either of the other two strategies.

Figure 6: The Multiclass Hump Simulation. We show the results of the multiclass trunk problem, as the number of classes increases from 2 to 10, with the number of dimensions and the number of samples fixed. Effect size is measured with Cohen's Kappa. LOL and PLS provide better performance over competing techniques including PCA, and this gap widens as the number of classes increases.

This paper also discounts many dimensionality reduction methods and investigates only PCA and random projection methods. While I agree with the paper that many dimensionality reduction methods do not meet the goals stated by this paper, I do believe that some further exploration would be useful to more appropriately place this paper within the literature. For instance, I am unconvinced that supervised autoencoders should be entirely discounted. While I agree that deep non-linear SAEs are uninterpretable black-box methods with too many tunable parameters, I wonder how a fully linear SAE with only a single hidden layer would perform compared to PCA/LOL. In fact, such an AE (without the supervised component) has been shown to be the same function class as PCA, so how does the corresponding SAE compare to

LOL? I would argue that the interpretability of this SAE is no worse (or better) than the interpretability of LOL.

Similar to the discussion on supervised autoencoders, I would have liked to see further discussion on supervised dictionary learning (the seminal SDL paper being cited by this paper, but not discussed). Extending SDL to out-of-sample prediction has been widely investigated and the goal of SDL is to directly parameterize the "projection" matrix that this paper seeks to hand-design. While the theory in this paper suggests that the hand-designed projection matrix is better than PCA, I wonder if it is likewise better than a direct parameterization/optimization using SDL (note this relationship can likewise be shown with SAEs, though it is obscured).

Paper describing SAEs for out-of-sample supervised classification (for transparency, note I am an author on this paper):

Le, Lei, Andrew Patterson, and Martha White. "Supervised autoencoders: Improving generalization performance with unsupervised regularizers." *Advances in Neural Information Processing Systems*. 2018.

Paper describing SDL for out-of-sample supervised classification:

Mairal, Julien, Francis Bach, and Jean Ponce. "Task-driven dictionary learning." *IEEE transactions on pattern analysis and machine intelligence* 34.4 (2011): 791-804.

Thank you for your detailed clarification regarding SAEs and SDLs. We agree that devising a unified strategy with respect to direct estimation (e.g., PCA, LOL) compared to optimization-based techniques (e.g., SDLs, SAEs) presents an excellent future direction for the work, and have reflected our discussion paragraph to highlight the promise these strategies might provide.

Recently discussed strategies have identified techniques for recovering the loading vectors for PCA from the weights of a single-layer auto-encoder [63]. Similarly, we believe there may be promise in devising a harmony between LOL and supervised auto-encoder strategies to identify the spaces spanned by the first and second moments in unison. Successive techniques, such as unsupervised regularization of supervised autoencoders [2] and supervised dictionary learners [1], may show promise for constructive development of the projection matrix through optimization, rather than estimation, techniques. Unfortunately, these approaches lack standard numerical packages for direct comparison, evaluation, and implementation. Future work may seek to highlight the similarities, or differences, possible through such techniques.

Minor comments/question

LOL can be seen as making two modifications to PCA. It uses the class-conditional covariance matrix to choose the d highest variance eigenvectors, then it also replaces $C-1$ of those d eigenvectors with the difference between class conditional means. How do either of these modifications perform in isolation? Based on my understanding of Theorem 1, using the class conditional means is the most important novelty, but does that hold in practice as well?

Thank you for this comment. We have clarified our text regarding the simulations in Figure 3 to clarify that the first $C-1$ dimensions of an LOL projection is equivalent to use of the class-conditional means in isolation. Further, the embedding technique we term "LDA" is really reduced-rank LDA as noted in "The Geometric Intuition of LOL". We have augmented our figures to be more specific about when we mean reduced-rank LDA and not the LDA classifier by renaming the LDA embedding "rrLDA". Indeed, rrLDA is equivalent to projection using only the class-conditional covariance matrix. Together, these two facts indicate the performance of these augmentations to PCA in isolation.

Figure 3 shows a two-dimensional scatterplot (left) and misclassification rate versus dimensionality (right) for each simulation. The top $C - 1$ embedding dimensions for LOL correspond to the performance after projection onto the class-conditional means, and rrLDA corresponds to the performance of projection onto the class-conditional covariance matrix. Figure 3A shows a three class generalization

In Figure 3, why does only LOL have a "robust" version? The performance gain of LOL in Figure 3b is negligible and much worse than many competitors, but "robust" LOL does quite well. Would using a robust PCA have the same effect? Or can only LOL be made robust in this way? The robust LOL method is never described that I could find it. My assumption is that you replaced the class-conditional mean with a class-conditional median. Likewise in Figure 3c, why not compare QOQ with QDA as well as QDA+PCA? Would this significantly outperform QOQ? Given that LDA and PCA both outperform LOL, I would expect that the quadratic versions would likewise outperform the quadratic version of LOL.

If this is true, then that means LOL only outperformed its baseline in one of three simulated settings and only for small values of d in that one setting.

We appreciate this feedback. The purpose of this figure was not to show simulations in which LOL itself always works; rather, the purpose of this figure was to show that a generalization of the XOX framework would readily apply, depending on how the data presents, based on relationships that exist within the data (such as whether there are outliers column 2, or whether there is an disparate covariance per-class column 3). Class-conditional moments simply allow one to capture these increasingly less simple relationships rather elegantly and simply into a unified projection technique. We have clarified this fact in the text.

These three results demonstrate how straightforward generalizations of LOL under the XOX framework which incorporate alternate or robust moment estimates can dramatically improve performance over other projection methods. This is in marked contrast to other approaches, for which such flexibility is either not available, or otherwise problematic.

In Figure 4, why can't PCA use the randomized approximate algorithm for eigendecomposition? Would PCA not also be able to benefit greatly in terms of computational performance, especially considering that LOL is simply PCA with a class-conditional mean centering preprocessing step (e.g. LOL and PCA rely on the exact same eigendecomposition code, LOL just has a subtraction step first).

Thank you for this point. Our purpose here was not to show that LOL with a randomized eigendecomposition (LFL) was superior performance and efficiency wise to PCA; it was merely to highlight that LFL provides an efficiency improvement over LOL, while maintaining the enhancement in performance over PCA. We have better highlighted the aims of this figure in the caption and discussion of Figure 4 in the main text.

Figure 4: Computational efficiency and scalability of LOL using $n = 2000$ samples from spherically symmetric Gaussian data (see Appendix C for details). **(A)** LOL exhibits optimal (linear) scale up, requiring only 46 minutes to find the projection on a 500 gigabyte dataset, and only 3 minutes using LFL (dashed lines show semi-external memory performance). **(B)** Error for LFL is the same as LOL in this setting, and both are significantly better than PCA and rrLDA for all choices of projection dimension, **regardless of whether a randomized approach is used to compute the projection dimensions.**

Finally, note that this simulation setting is ideal for PCA and rrLDA, because the first principal component includes the mean difference vector. Nonetheless, both LOL and LFL achieve near optimal accuracy, whereas rrLDA is at chance, and PCA requires 500 dimensions to even approach the same accuracy that LOL achieves with only one dimension. **While PCA would also benefit efficiency wise from a randomized approach, we emphasize that LFL maintains the high performance of LOL in comparison to PCA despite the randomization technique, with the benefit of greater computational efficiency compared to LOL.**

What is Figure 5 showing for the LDA columns? Since RR-LDA is equivalent to LDA composed with PCA, then it can't be showing the reduced rank form. I'm not sure that LDA can be composed with LDA, that doesn't make sense to me (thus the label is wrong, I believe), so what is the LDA column in Figure 5a and row in Figure 5b? Is it LDA applied to the raw dataset without dimensionality reduction?

We appreciate this point, and it appears our slight overloading of the term “LDA” has caused confusion. We have remedied this fact by replacing use of the term “LDA” when referring to the embedding technique with “rrLDA” throughout, which is the distinction between the embedding technique which uses the top d eigenvectors of the class-conditional covariance matrix (rrLDA) with the full classification technique (LDA, which is a classification technique which can be thought of as projecting the data using the class-conditional covariance matrix).

Note that the sample class-centered covariance matrix estimates the population covariance, colored- whereas the sample pooled covariance matrix is distorted by the difference of the class means. Further, as discussed in Appendix D, the class-centered covariance matrix is equivalent to “Reduced Rank LDA” [42] (**rrLDA** hereafter, which is simply LDA but truncating the covariance matrix). For the theoretical background on LDA and **rrLDA**, a formal definition of LOL, and detailed description of the simulation settings that follow, see Appendices A, B, and C, respectively. Figure 2 shows three different examples of 100 data points sampled from a 1,000 dimensional Gaussian to geometrically illustrate the intuition that motivated LOL. In each case, all dimensions are uncorrelated with one another, and all classes are equally likely with the same covariance; the only difference between the classes are their means.

In Figure 6, you show LOL used for a regression problem. This is quite interesting, and in general I would love to see more exploration of LOL used with other classification algorithms (other than LDA). However, how can LOL be used for regression? How do you compute the class-conditional covariance and means for a regression problem? Without these, I would expect LOL to revert back to PCA.

Thank you for this point. We have clarified in section F. to explicitly separate regression and hypothesis testing. Further, as stated in the description in F.II regression is performed by partitioning the data into K-partitions based on the percentile of the target variable; we have clarified that we selected K=10 arbitrarily.

F.III Regression

High-dimensional regression is another supervised learning method that can use the LOL idea. Linear regression, like classification and Hotelling’s Test, requires inverting a matrix as well. By projecting the data onto a lower-dimensional subspace first, followed by linear regression on the low-dimensional data, we can mitigate the curse of high-dimensions. To choose the projection matrix, we partition the data into K partitions (we select $K = 10$ arbitrarily), based on the percentile of the target variable, we obtain a K-class classification problem. Then, we can apply LOL to learn the projection. Figure 7C shows an example of this approach, contrasted with Lasso and partial least squares, in a sparse simulation setting (see Methods for details). LOL is able to find a better low-dimensional projection than Lasso, and performs significantly better than partial least squares, for essentially all choices of number of dimensions to project into.

I believe appendix B has a few typos:

- B.II: $g^d_{\{LDA\}}$ is defined twice. I believe the first definition should be labelled $g_{\{LDA\}}$
- B.III: δ is defined as a matrix of size $C-2 * p$ instead of $C-1 * p$. Should the last index in this array be δ_C instead of $\delta_{\{C-1\}}$?
- B.III: $A^D_{\{LOL\}}$ is defined twice and in two different ways. The first definition assumes two-class and the second definition is more general for any number of C.

Thank you for pointing out these typos. We have clarified accordingly.

Andrew Patterson

Reviewer #3 (Remarks to the Author):

Summary:

The paper discusses the use of linear projection (PCA) per class followed by LDA to preserve the discriminatory power of features. The LOL method ensures that the projected data is useful for the subsequent linear classification problem.

Comments:

Unfortunately, the idea of appending the mean difference vector to features and class-centering to convert unsupervised manifold learning to a supervised manifold learning setting are not a substantial enough contribution to ML methodology. The claims of scalability are not interesting either.

While whether an advance is interesting is subjective, we find the scalability to be interesting; particularly, we have been unable to find other tools that run on datasets with billions of features. If there are papers addressing similar scalability performance, we would be very interested to try them out.

The draft has not been proofread properly has elementary mistakes even in the background/preliminary sections such as A.I and A.II.

We appreciate you pointing out the typos in the appendix; we have gone through the entire appendix section and attempted to eliminate the ones noted (as well as additional typos that were found).

>> Existing linear and nonlinear dimensionality reduction methods either are not supervised, scale poorly to operate in big data regimes, lack theoretical guarantees, or are “black-box” methods unsuitable for many applications.

This is an exaggerated mis-characterization of work in this area. There are multiple papers that address scalability, address theory of such dimensionality reduction, as well as are quite interpretable and transparent.

We agree our initial claim was too strong, and have amended our wording to reflect that this tends to be characteristic of work in the area, and is not necessarily the case for all existing linear and nonlinear dimensionality reduction techniques. Further, we have specified explicitly a focus on deep learning methods with respect to the specific wording of this paragraph, which was what we were trying to focus on initially.

Most manifold learning methods, while exhibiting both strong theoretical [6, 29, 33] and empirical performance, are typically fully unsupervised. Thus, in classification problems, they discover a low-dimensional representation of the data, ignoring the labels. This approach can be highly problematic when the discriminant dimensions and the directions of maximal variance in the learned manifold are not aligned (see Figure 1 for some examples). Moreover, nonlinear manifold learning techniques tend to learn a mapping from the original samples to a low-dimensional space, but do not learn a projection, meaning that new samples cannot easily be mapped onto the low-dimensional space, a requirement for supervised learning. **Deep learning methods [38] can easily be supervised, but they tend to require huge sample sizes, lack theoretical guarantees, or are opaque “black-boxes” that are insufficient for many biomedical applications. This yields a dearth of “out of the box” supervised scalable dimensionality reduction techniques with strong theoretical guarantees with respect to classification performance bounds designed for wide datasets.** Random forests circumvent many of these problems, but implementations that operate on millions of dimensions do not exist [70], **and often produce embeddings that perform no better than PCA on wide datasets (Figure 5).**

>> LOL achieves achieves state-of-the-art classification accuracy

This seems to be an exaggerated claim. Partly because restricting oneself to linear classification settings already limits the model representation power. What does state of the art even mean here?

We agree this claim was exaggerated as well, and have modified the wording in the abstract to be more specific regarding LOL outperforming readily scalable, linear, dimensionality reduction techniques, which are the prime focus of most of our comparisons.

To solve key biomedical problems, experimentalists now routinely measure millions or billions of features (dimensions) per sample, with the hope that data science techniques will be able to build accurate data-driven inferences. Because sample sizes are typically orders of magnitude smaller than the dimensionality of these data, valid inferences require finding a low-dimensional representation that preserves the discriminating information (e.g., whether the individual suffers from a particular disease). There is a lack of interpretable supervised dimensionality reduction methods that scale to millions of dimensions with strong statistical theoretical guarantees. We introduce an approach, XOX, to extending principal components analysis by incorporating class-conditional moment estimates into the low-dimensional projection. The simplest version, “Linear Optimal Low-rank” projection (LOL), incorporates the class-conditional means. We prove, and substantiate with both synthetic and real data benchmarks, that LOL and its generalizations in the XOX framework lead to improved data representations for subsequent classification, while maintaining computational efficiency and scalability. Using multiple brain imaging datasets consisting of >150 million features, and several genomics datasets with >500,000 features, LOL outperforms other scalable linear dimensionality reduction techniques in terms of accuracy, while only requiring a few minutes on a standard desktop computer.

Additional Comments:

- The first expression for $g_{\text{LDA}}(\mathbf{x})$ in pg 19 looks wrong. The first term should also be a function of y .

- Why did the narrative change to two class instead of the general setting in A.II pg 19 even before calling out for the two class setting (i.e., why $\pi_0 = \pi_1$ before talking about the two-class setting)?

Thank you for the above two comments; we have rewritten this section to be more clear about what is happening, and fixed a typo regarding $g_{\text{LDA}}(\mathbf{x})$.

A.II Linear Discriminant Analysis (LDA)

Linear Discriminant Analysis (LDA) is an approach to classification that uses a linear function of the first two moments of the distribution of the data. More specifically, let $\boldsymbol{\mu}_j = \mathbb{E}[F_{X|Y=j}]$ denote the class conditional mean, and let $\boldsymbol{\Sigma} = \mathbb{E}[F_X^2]$ denote the joint covariance matrix, and the class priors are $\pi_j = \mathbb{P}[Y = j]$. Using this notation, we can define the LDA classifier:

$$g_{\text{LDA}}(\mathbf{x}) := \underset{y}{\operatorname{argmin}} \left[\frac{1}{2}(\mathbf{x} - \boldsymbol{\mu}_y)^\top \boldsymbol{\Sigma}^{-1}(\mathbf{x} - \boldsymbol{\mu}_y) - \log \pi_y \right],$$

Let L_{LDA}^F be the expected misclassification rate of the above classifier for distribution F . Assuming equal class prior and centered means, re-arranging a bit, we obtain

$$g_{\text{LDA}}(\mathbf{x}) := \underset{y}{\operatorname{argmin}} \mathbf{x}^\top \boldsymbol{\Sigma}^{-1} \boldsymbol{\mu}_y.$$

In words, the LDA classifier chooses the class that maximizes the magnitude of the projection of an input vector \mathbf{x} onto $\boldsymbol{\Sigma}^{-1} \boldsymbol{\mu}_y$. When there are only two classes, letting $\boldsymbol{\delta} = \boldsymbol{\mu}_0 - \boldsymbol{\mu}_1$, the above further simplifies to

$$g_{2\text{LDA}}(\mathbf{x}) := \mathbb{I}\{\mathbf{x}^\top \boldsymbol{\Sigma}^{-1} \boldsymbol{\delta} > 0\}.$$

Note that the equal class prior and centered means assumptions merely changes the threshold constant from 0 to some other constant.

- Lemma 1 is trivial/not new.

- Eq (2) pg 20 is not new.

Thank you for this comment; we agree, we were merely providing them for reference for interested readers who were unfamiliar with them. We have since deleted the one-line proof and replaced it with a reference.

- What is the difference between L_{A}^d and L_{A} ? Please be consistent in notation.

Thank you for this comment; we had forgotten the superscript d previously inside the integral.

Let $L_A^d := \int \mathbb{P}[g_A^d(\mathbf{x}) = y] f_{\mathbf{x},y} d\mathbf{x}dy$. Our goal therefore is to be able to choose A for a given parameter setting $\theta = (\pi, \delta, \Sigma)$, such that L_A is as small as possible (note that L_A will never be smaller than L_*).

- In Eq (3), is Σ and δ (which determine Σ_A and δ_A) already known? This should have been clarified as this is very important and makes the optimization problem difficult/easy.

Why is this a deal breaker. There are many problems where the population optimal estimation or model is not computable? Also, can you comment on what type of problem this is? non-convex? for what additional assumptions could it become convex?

We appreciate this feedback, and have adjusted the text regarding this statement to clarify that we are specifically referring to the case where (Σ, δ) are not known a priori. The problem becomes difficult due to its non-convexity. Further, we cannot evaluate the integral directly. One could solve a convex *approximation* through various sets of assumptions (such as assuming that Σ or δ are known); we do not.

In the naive case where (Σ_A, δ_A) are known, we seek to solve the following linear optimization problem:

$$\begin{aligned} & \underset{A}{\text{minimize}} && \mathbb{E}[\mathbb{I}\{\mathbf{x}^T \mathbf{A}^T \Sigma_A^{-1} \delta_A > 0\} \neq y] \\ & \text{subject to} && A \in \mathbb{R}^{d \times p}. \end{aligned} \tag{3}$$

When (Σ_A, δ_A) are not known, however, the optimization problem becomes non-convex. With Σ_A and δ_A as above:

$$\begin{aligned} & \underset{A, \Sigma, \delta}{\text{minimize}} && \mathbb{E}[\mathbb{I}\{\mathbf{x}^T \mathbf{A}^T \Sigma_A^{-1} \delta_A > 0\} \neq y] \\ & \text{subject to} && A \in \mathbb{R}^{d \times p}. \end{aligned} \tag{4}$$

While there are numerous approaches to solve related convex optimization problems through various sets of assumptions [34, 57], we do not consider such techniques in this manuscript theoretically. This is because assuming either a structure for Σ_A or δ_A presupposes an understanding of the properties of the feature space for wide data, which is often unsuitable if the dataset is large or has considerable complexity.

- What is \mathcal{A}^d and \mathcal{A} ? If they are the same, then how can $\delta^T \Sigma^{-1}$ belong to this set. The dimensions don't match, so Lemma 2 is incorrect.

Thank you for this comment; this was a typo; \mathcal{A} is the set of possible embeddings into $\leq p$ dimensions, whereas \mathcal{A}^d is the set of possible embeddings into d dimensions.

Let $\mathcal{A} = \{A : A \in \mathbb{R}^{k \times p}, k \leq p\}$, $\mathcal{A}^d = \{A : A \in \mathbb{R}^{d \times p}\}$, and let $\mathcal{A}_* \subset \mathcal{A}$ be the set of A that minimizes Eq. (4), and let $A_* \in \mathcal{A}_*$. Let $L_A^* = L_{A_*}$ be the misclassification rate for any $A \in \mathcal{A}_*$, that is, L_A^* is the Bayes optimal misclassification rate for the classifier that composes A with LDA.

- Lemma 2 proof: what is δ_B ?

We have clarified the definition accordingly, by plugging in B from the definition given in the preceding paragraph:

Lemma 2. $\delta^T \Sigma^{-1} \in \mathcal{A}_*$

Proof. Let $B = (\Sigma^{-1} \delta)^T = \delta^T (\Sigma^{-1})^T = \delta^T \Sigma^{-1}$, so that $B^T = \Sigma^{-1} \delta$, and plugging this in to Eq. (2).
By the above, and noting the symmetry and invertibility of Σ :

$$\begin{aligned}\Sigma_B &= B \Sigma B^T = \delta^T \Sigma^{-1} \Sigma (\delta^T \Sigma^{-1})^T \\ &= \delta^T \Sigma^{-1} \Sigma \Sigma^{-1} \delta = \delta^T \Sigma^{-1} \delta \\ \Rightarrow \Sigma_B^{-1} &= \delta^{-1} \Sigma \delta^T \\ \delta_B &= B \delta = \delta^T \Sigma^{-1} \delta\end{aligned}$$

We obtain:

$$\begin{aligned}g_B(x) &= \mathbb{I}\{x^T B^T \Sigma_B^{-1} \delta_B > 0\} \\ &= \mathbb{I}\{x^T (\Sigma^{-1} \delta) (\Sigma_B^{-1} \delta_B) > 0\} && \text{plugging in } B \\ &= \mathbb{I}\{x^T (\Sigma^{-1} \delta) (\delta^{-1} \Sigma \delta^T \Sigma^{-1} \delta) > 0\} && \text{plug in } \Sigma_B, \delta_B \text{ from above} \\ &= \mathbb{I}\{x^T \Sigma^{-1} \delta > 0\}\end{aligned}$$

In other words, letting B be the Bayes optimal projection recovers the Bayes classifier, as it should.
Or, more formally, for any $F \in \mathcal{F}_{\text{LDA}}$, $L_{\delta^T \Sigma^{-1}} = L_*$. □

REVIEWER COMMENTS

Reviewer #1 (Remarks to the Author):

I thank the authors for addressing my comments. I am satisfied with the revision in general. I have two minor comments:

1. In the response, the authors mentioned that the median is a "robust estimator of the first moment". I don't think the statement is correct. Median and mean are fundamentally different. When the distribution is asymmetric, the sample median is not a consistent estimator of the population mean. There have been intensive study on robustifying the mean and covariance estimation, e.g., [1], [2], [3].

2. In Figure 3, it seems to me that LOL or QOQ does not significantly outperform PCA after tuning of embedding dimensions. Please comment on this.

References:

[1] Fan, J., Wang, W. and Zhu, Z. A Shrinkage Principle for Heavy-Tailed Data: High-Dimensional Robust Low-Rank Matrix Recovery.

[2] Ke, Y., Minsker, S., Ren, Z., Sun, Q., Zhou, W. User-Friendly Covariance Estimation for Heavy-Tailed Distributions.

[3] Wei, X., Minsker, S. Estimation of the covariance structure of heavy-tailed distributions.

Reviewer #2 (Remarks to the Author):

After reading the author response and edits to the paper, as well as the conversation between the authors and other reviewers, I am happy with state of the paper and would recommend publication in this venue. There are a few minor modifications that I would encourage the authors to make with regards to wording.

1) The robust version of LOL (rLOL) does not make use of class-conditional *moment* estimates, but rather class-conditional *statistic* estimates (i.e. the median is not a moment). A nitpick to be sure, but I think this makes the method a little more clear while also hinting towards greater generalization of the method.

2) I disagree with the reasoning provided for why SAEs and SDLs are not used. That a standard numerical package does not exist does not preclude the need to compare to these methods, especially given that they are trivial to implement using a standard optimization package (of which there are many). That said, on reconsidering the intended audience of this paper, I think it is not strictly necessary to compare to SAEs and SDLs. I would remove the citation to the SAE paper, I'm not sure that it adds value and is likely not the best citation for SAEs (just the first that came to my mind for obvious reasons!), I simply brought it up to clarify my point.

3) The left subplot of figure 4 directly compares the runtime of LFL to PCA. If you included a similar enhancement to PCA, then the LFL+PCA line would lie directly on top of the current

LFL line (much like how LOL lies on top of PCA currently). This subplot is at best meaningless and at worst misleading. If you make the comparison strictly fair, then the plot no longer shows a meaningful relationship between LOL and PCA and if you leave it as is, then the clear naive interpretation is that LOL is better than PCA because it can be made more efficient which is misleading. I think the best course of action would be to remove this subplot and simply leave the textual description. Something like: "LOL requires 46 minutes to find the projection ... where LFL requires only 3 minutes. Note similar performance enhancements can be made to PCA and here our focus is to highlight that LFL maintains the high performance of LOL in comparison to PCA despite the randomization technique."

Andrew Patterson

The point-by-point is organized as follows. Comments from reviewers are enumerated in black text. Our responses are provided in blue text. Relevant text changes which specifically address the reviewer's comment are indicated by the red text within the screenshot (and the corresponding marked up manuscript file, lol (13).pdf, which differs from the LaTeX file provided in that it has changes reflected in red).

Reviewer #1 (Remarks to the Author):

I thank the authors for addressing my comments. I am satisfied with the revision in general. I have two minor comments:

1. In the response, the authors mentioned that the median is a "robust estimator of the first moment". I don't think the statement is correct. Median and mean are fundamentally different. When the distribution is asymmetric, the sample median is not a consistent estimator of the population mean. There have been intensive study on robustifying the mean and covariance estimation, e.g., [1], [2], [3].

Thank you for pointing out this error. We have updated our description of LOL to clarify that one can use either moments or robust measures of location and scale, and have updated our description from "robust estimator of the first moment" to a "robust measure of location". Further, the citations for robust means you provide are excellent, and we have added those as well.

Figure 3 shows a two-dimensional scatterplot (left) and misclassification rate versus dimensionality (right) for each simulation. Hereafter, LOL will refer to the version of LOL with a **robust estimate of the location** (the class medians, related to the central moment when the population has a symmetric distribution), and a truncated singular value decomposition to estimate of the second moment. A robust location estimate tends to make little difference when a robust estimate was not necessary, and empirically improves performance in simulations and real-data examples when a robust estimate was warranted. Alternative strategies would have been to use robust estimates of the first moment or second moment directly [32, 47, 61]. We do not use a robust estimate of the second moment, as typical robust estimates of the second moment available in standard numerical packages require $d < n$, which is unsuitable for wide data. The top $C - 1$ embedding dimensions for LOL correspond to the perfor-

2. In Figure 3, it seems to me that LOL or QOQ does not significantly outperform PCA after tuning of embedding dimensions. Please comment on this.

This situation was designed to be extremely troublesome to XOX algorithms which use a component incorporating the first moment in the leading dimensions. We have added a much more in-depth discussion as to what is going on here, included below. Effectively, both LOL and QOQ are incorporating useless information (the difference of the means), but still do not do much worse than PCA (and in fact, QOQ does better than PCA, despite incorporating uninformative information).

Figure 3C shows an example which should be adversarial for LOL in comparison to PCA or rrLDA. This is because the difference of means is utterly informative, so LOL utilizes additional dimensions which are noise compared to PCA. Further, the class-conditional covariances are orthogonal, whereas LOL assumes the class-conditional covariance is the same across both classes. While LOL cannot possibly do as well as PCA in this situation, its performance is only slightly worse. Further, another XOX variant, Quadratic Optimal QDA (QOQ), uses the same difference of means as LOL and then computes the eigenvectors separately for each class, concatenates them (sorting them according to their singular values), and then classifies with QDA instead of LDA. QOQ is able to identify a slightly more efficient projection for classification than PCA. This is due to the fact that while the first few dimensions are uninformative (those spanned by the difference of the means), the successive dimensions are far

7

more efficient (the class-conditional covariances).

References:

- [1] Fan, J., Wang, W. and Zhu, Z. A Shrinkage Principle for Heavy-Tailed Data: High-Dimensional Robust Low-Rank Matrix Recovery.
- [2] Ke, Y., Minsker, S., Ren, Z., Sun, Q., Zhou, W. User-Friendly Covariance Estimation for Heavy-Tailed Distributions.
- [3] Wei, X., Minsker, S. Estimation of the covariance structure of heavy-tailed distributions.

Reviewer #2 (Remarks to the Author):

After reading the author response and edits to the paper, as well as the conversation between the authors and other reviewers, I am happy with state of the paper and would recommend publication in this venue. There are a few minor modifications that I would encourage the authors to make with regards to wording.

1) The robust version of LOL (rLOL) does not make use of class-conditional *moment* estimates, but rather class-conditional *statistic* estimates (i.e. the median is not a moment). A nitpick to be sure, but I think this makes the method a little more clear while also hinting towards greater generalization of the method.

Thank you very much for this feedback. As mentioned in the response to the previous reviewer, we have clarified that in practice, while the theory behind LOL applies to moments, in practice we can readily generalize the framework to other statistics (such as measures of scale and location, in place of moments).

2) I disagree with the reasoning provided for why SAEs and SDLs are not used. That a standard numerical package does not exist does not preclude the need to compare to these methods, especially given that they are trivial to implement using a standard optimization package (of which there are many). That said, on reconsidering the intended audience of this paper, I think it is not strictly necessary to compare to SAEs and SDLs. I would remove the citation to the SAE paper, I'm not sure that it adds value and is likely not the best citation for SAEs (just the first that came to my mind for obvious reasons!), I simply brought it up to clarify my point.

We have removed the corresponding discussion regarding SAEs.

3) The left subplot of figure 4 directly compares the runtime of LFL to PCA. If you included a similar enhancement to PCA, then the LFL+PCA line would lie directly on top of the current LFL line (much like how LOL lies on top of PCA currently). This subplot is at best meaningless and at worst misleading. If you make the comparison strictly fair, then the plot no longer shows a meaningful relationship between LOL and PCA and if you leave it as is, then the clear naive interpretation is that LOL is better than PCA because it can be made more efficient which is misleading. I think the best course of action would be to remove this subplot and simply leave the textual description. Something like: "LOL requires 46 minutes to find the projection ... where LFL requires only 3 minutes. Note similar performance enhancements can be made to PCA and here our focus is to highlight that LFL maintains the high performance of LOL in comparison to PCA despite the randomization technique."

Thank you for this feedback. We have modified the figure caption to add a description.

Figure 4: Computational efficiency and scalability of LOL using $n = 2000$ samples from spherically symmetric Gaussian data (see Appendix C for details). **(A)** LOL exhibits optimal (linear) scale up, requiring only 46 minutes to find the projection on a 500 gigabyte dataset, and only 3 minutes using LFL (dashed lines show semi-external memory performance). **(B)** Error for LFL is the same as LOL in this setting, and both are significantly better than PCA and rrLDA for all choices of projection dimension, regardless of whether a randomized approach is used to compute the projection dimensions. Note that while similar scalability enhancements can be made to PCA in (A), our focus is to highlight that LFL maintains the high performance of LOL in comparison to PCA in (B) despite the randomization technique.